# Essential Oils from Colombian Plants: Antiviral Potential against Dengue Virus Based on Chemical Composition, In Vitro and In Silico Analyses

**DOI:** 10.3390/molecules27206844

**Published:** 2022-10-12

**Authors:** Lina Silva-Trujillo, Elizabeth Quintero-Rueda, Elena E. Stashenko, Sergio Conde-Ocazionez, Paola Rondón-Villarreal, Raquel E. Ocazionez

**Affiliations:** 1Centro de Cromatografía y Espectrometría de Masas, CROM-MASS, Universidad Industrial de Santander, Bucaramanga 680002, Santander, Colombia; 2Instituto de Investigación Masira, Facultad de Ciencias de la Salud, Universidad de Santander, Bucaramanga 680003, Santander, Colombia

**Keywords:** dengue, dengue virus, essential oils, terpenes

## Abstract

Currently, there are no therapies to prevent severe dengue disease. Essential oils (EOs) can serve as primary sources for research and the discovery of phytomedicines for alternative therapy. Fourteen EOs samples were obtained by distillation from six plants used in Colombian folk medicine. GC/MS analysis identified 125 terpenes. Cytopathic effect (CPE) reduction assays revealed differences in antiviral activity. EOs of *Lippia alba*, citral chemotype and carvone-rich fraction; *Lippia origanoides*, phellandrene chemotype; and *Turnera diffusa*, exhibited strong antiviral activity (IC_50_: 29 to 82 µg/mL; SI: 5.5 to 14.3). EOs of *Piper aduncum*, *Ocimum basilicum*, and *L. origanoides*, carvacrol, and thymol chemotypes, exhibited weak antiviral activity (32 to 53% DENV-CPE reduction at 100 µg/mL; SI > 5.0). Cluster and one-way ANOVA analyses suggest that the strong antiviral activity of EOs could be attributed to increased amounts of non-phenolic oxygenated monoterpenes and sesquiterpene hydrocarbons. Docking analyses (AutoDock Vina) predicted binding affinity between the DENV-2 E protein and terpenes: twenty sesquiterpene hydrocarbons (−8.73 to −6.91 kcal/mol), eight oxygenated monoterpenes (−7.52 to −6.98 kcal/mol), and seven monoterpene hydrocarbons (−7.60 to −6.99 kcal/mol). This study reports for the first time differences in the antiviral activity of EOs against DENV, corresponding to their composition of monoterpenes and sesquiterpenes.

## 1. Introduction

Dengue virus (DENV) is transmitted to humans by infected *Aedes* mosquitoes. The virus is prevalent in more than 100 countries worldwide [1]. DENV infections manifest in a broad spectrum of presentations, including asymptomatic infection, mild flu-like syndrome, and severe disease [2]. Severe dengue is a life-threatening worsening of dengue symptoms, which remains one of the leading causes of hospitalization in developing and underdeveloped countries, where the surveillance network for disease control is not robust [3]. There are no effective drugs to prevent the development of severe dengue, although intensive research on synthetic antivirals has been ongoing for decades [4,5]. Dengue is a complex disease, which has made the discovery of effective therapies difficult. Studies support the view that herbal medication could be useful for treating DENV infections, which could reduce the risk of severe dengue if started soon after exposure to the virus [6,7,8]. 

Essential oils (EOs) distilled from aromatic plants have traditionally been used to prepare medicinal herbs to treat human diseases [9,10]. Numerous EOs exhibit in vitro antiviral activity against pathogenic human-enveloped viruses such as herpesvirus, coronavirus, influenza virus, and human immunodeficiency virus [11,12,13]. EOs are also recognized for their anti-inflammatory and stimulatory effects on the human immune system [14,15]. Monoterpenes and sesquiterpenes are the main constituents of EOs and have been considered their active components against pathogenic viruses due to their antiviral and anti-inflammatory properties [9,10]. EOs have been proposed as alternative medicines for the treatment of various antiviral diseases [12,13]. Recently, EOs with established pharmacokinetic and pharmacodynamic properties were proposed as an alternative treatment to prevent clinical complications associated with COVID-19 [16,17]. 

Dengue is a serious health problem in Colombia. In the last decade, epidemics have occurred with high mortality rates, compared to the average values reported in Latin American countries [3]. Despite the richness of Colombian flora, characterized by more than 30,000 species of higher plants [18], only a small number of species have been studied to discover their potential as primary sources of herbal medicines. Research focusing on the antiviral potential of EOs may contribute to the discovery of alternative therapies to prevent severe dengue.

Plants used in folk medicine in Colombia and Latin American countries were selected for the present study. *Lippia alba* (Mill.) N.E.Br. ex Britt and Wils (Verbenaceae) is used as a remedy for treating stomach disorders, influenza, and headaches [19]. *Lippia origanoides* Kunth (Verbenaceae) is employed to prepare gastrointestinal and respiratory remedies [20]. *Turnera diffusa* Willd ex Schult (Passifloraceae), commonly known as damiana, is used as a tonic and sexual stimulant, and to treat influenza, gastrointestinal and skin disorders [21]. *Piper aduncum* L. (Piperaceae) is utilized in traditional medicine for its antimicrobial, anti-inflammatory, anthelminthic, and analgesic properties [22]. *Varronia curassavica* Jacq. (Boraginaceae) is used to treat inflammation, ulcers, arthritis, and pain; its EO is employed to treat myofascial pain and tendonitis [23]. *Ocimum basilicum* L. (Lamiaceae) has extensive applications in the culinary, cosmetics, nutraceutical, and toiletry industries; its EO is utilized to alleviate mental fatigue, colds, rhinitis, and to treat snakebites [24]. 

Variations of the biological activities of EOs according to specie, chemotype, and phenological stages of the plant are documented [9,10,25]. Therefore, studies on the variation of the antiviral activity with respect to the chemical composition of EOs could provide information on what determines their effects on virus infectivity. In this study, fourteen EOs obtained from the above-mentioned medicinal plants grown in Colombia were analyzed to determine the variation of antiviral activities against two DENV serotypes and the influence of terpene composition. In addition, molecular docking analyses were performed to predict the interactions of EO chemical constituents with DENV structural proteins.

## 2. Results

### 2.1. EO Chemical Composition 

Fourteen EOs from six medicinal plants grown in Colombia were studied (Table 1). A total of 125 compounds were identified by GC/MS, most of which are terpenes. Relative amounts measured by GC/FID and the linear retention indices of compounds in order of their elution on the DB-5MS column are listed in Appendix A. The main constituents of all fourteen EOs are presented in Table 2. 

*L. origanoides* EOs had more monoterpenes (59.8 to 94.3%) than sesquiterpenes (2.5 to 35.3%). In the LoP (phellandrene chemotype) sample, *trans*-β-caryophyllene (15.1%), thymol (14%), 1,8-cineol (13%) and *p*-cymene (12.6%) were the main constituents followed by α-phellandrene (7.1%). In thymol chemotype EOs, thymol was the main constituent: 82.9, 75.3, and 49.4% in LoTf, LoT, and LoTC, respectively. In the LoC (carvacrol chemotype) sample, carvacrol (35%) and *p*-cymene (14.4%) were the main constituents.

*L. alba* EOs had a higher content of monoterpenes (66.2 to 99%) than sesquiterpenes (0.6 to 23.5%). In citral chemotype EOs, geranial and geraniol were the main constituents: LaCi, 24.5 and 19.0%; LaCif, 24.8 and 8.1%, respectively. In carvone chemotype EOs, limonene (82.2%) and carvone (78.2%) were the main constituents. 

Both *T. diffusa* EOs (TdS1 and TdS2) can be classified as sesquiterpene-rich EOs (86.2 to 87.2%), aristolochene (17.9 and 20.9%) and dehydrofukinone (25.4 and 19.3%) were their main constituents. 

*P. aduncum* EO (PaS1) was characterized by similar amounts of sesquiterpenes (46.9%) and monoterpenes (44.2%), piperitone (14.8%), and *trans*-β-caryophyllene (7.4%) were the main constituents. 

*O. basilicum* EO (ObS1) had a higher content of monoterpenes (72.3%) than sesquiterpenes (27.7%), linalool (42.7%) and estragol (18.6%) were the main constituents. 

*V. curassavica* EO (VcS1) was characterized by high (76.2%) content of sesquiterpenes, *trans*-β-caryophyllene (19.2%), germacrene D (12.3%), and *trans*-β-guaiene (11.8%) were the principal constituents. 

### 2.2. Antiviral Activity 

The crystal violet assay was used to evaluate the cytotoxicity of the EOs against Vero cells (Appendix A). No sample of EO reduced cell viability by more than 50%. Maximum non-cytotoxic concentration (MNTC) values were estimated to be in the range of 429 and 512 µg/mL. 

DENV-induced cytopathic effect (DENV CPE) is a surrogate measure of virus replication in vitro [29]. The lower the CPE, the higher the antiviral activity of the test sample. EOs were tested at six-point non-cytotoxic concentrations to evaluate antiviral activity against two serotypes of DENV (DENV-1 and DENV-2) during adsorption into the cell (Figure 1, Table 3). Four EO samples (LaCi and LaCaf2 of *L. alba*; LoP of *L. origanoides*; and TdS2 of *T. diffusa* 2016) reduced the CPE of both DENV serotypes in a dose-dependent manner. The IC_50_ values range from 29 to 82 µg/mL, and the selectivity indices were 5.5 to 14.3; therefore, these four EOs were classified as having a strong antiviral activity. Six EO samples (LaCaf1 of *L. alba*; LoC, LoTC, and LoTf of *L. origanoides;* ObS1 of *O. basilicum*; and PaS1 of *P. aduncum*) reduced by 32–53% (one-way ANOVA, *p*: < 0.0001) the CPE of both DENV serotypes at the highest concentration of 100 µg/mL, but not at the other oil concentrations. Therefore, IC_50_ values could not be estimated, and these six EOs were classified as having weak antiviral activity. The remaining four EOs (TdS1, LaCif, LoT, and VcS1) were classified as inactive. DENV-CPE (%) of both virus serotypes was not different from the untreated control at all six concentrations of the EOs. 

Essential oils are described in Table 1. The IC_50_ (µg/mL) values are the means (±SD) of six measurements from three independent assays. Selectivity index (SI): MNTC in the cytotoxic assay/IC_50_. Strong activity, reduction of DENV CPE in EO-treated cells in a dose-dependent manner. Weak activity, reduction of DENV CPE in EO-treated cells at 100 µg/mL (DENV CPE: 32 to 53%, respect to 100% untreated control; one-way ANOVA: F: 48–1295, *p* < 0.001; Dunnett’s post hoc test, *p* < 0.001), but not at other concentrations. Inactive, there was no reduction of DENV CPE in EO-treated cells at the six concentrations tested.

Comparisons of antiviral activities against DENV-2 and terpene content of the EOs are described below: *L. origanoides* EOs (Figure 2A). LoP (phellandrene chemotype), with strong antiviral activity, had a higher content of sesquiterpene hydrocarbons (30.2%) and oxygenated monoterpenes (alcohols and oxides: 15.1%) compared to the EOs with weak antiviral activity (LoC, carvacrol chemotype; LoTC, thymol-carvacrol chemotype; and LoTf, thymol chemotype, fraction: 2.5 to 10.6% sesquiterpenes and 0 to 3.1% monoterpenes). Conversely, LoP had lower content (15.7%) of phenolic monoterpenes than LoTf (84.1%), LoC (46.9%), and LoTC (54.4%). The LoT EO (thymol chemotype), inactive against both virus serotypes, had the lowest amounts of monoterpene hydrocarbons among the five EO samples (3.6 *vs* 8.9%, in LoTf to 41.1% in LoTC). 

*L. alba* EOs *(*Figure 2B). LaCi (citral chemotype) and LaCaf2 (carvone-rich fraction), with strong antiviral activity, had higher amounts of oxygenated monoterpenes (alcohols, aldehydes, and ketones: 63.4% and 91.7%, respectively) compared to LaCaf1 (limonenerich fraction, 14.1%) with weak antiviral activity. On the other hand, the LaCif sample EO (citral, light fraction), which was inactive against both virus serotypes, had lower amounts of monoterpene alcohols (12.8%) and sesquiterpene oxides (0.1%) compared to LaCi (22.6% monoterpenes and 1.8% sesquiterpenes). 

*T. diffusa* EOs (Figure 3A). TdS2, with strong antiviral activity, but not TdS1, without antiviral activity, had monoterpene alcohols (0.3%) and phenolic monoterpenes (0.6%). Also, TdS2 had a higher content of sesquiterpene hydrocarbons (52.6%) than TdS1 (45.6%)

*P. aduncum* (PaS1), *O. basilicum* (ObS1)*,* and *V. curassavica* (VcS1) EOs (Figure 3B): PaS1 and ObS1, which exhibited weak antiviral activity, had higher amounts (20.1 and 69.4%, respectively of oxygenated monoterpenes (alcohols, ethers, and oxides) compared to the *V. curassavica* EO (0.8%), which was inactive against both virus serotypes. 

### 2.3. Chemical Cluster of EOs and Antiviral Activity

The relationship between antiviral activity and chemical composition was explored to test the hypothesis that, for those EOs showing strong and weak antiviral activity (CPE ≤ 60%, *n* = 10, Figure 4A), similar terpene content results in similar antiviral activity. Kohonen’s self-organized maps algorithm, using the amounts of terpenes exclusively as input (Figure 2 and Figure 3), grouped EOs in three different clusters (Figure 4B,C). 

Cluster 1 grouped three EOs (LaCi, LaCaf2, and TdS2) with strong antiviral activity and one EO (ObS1) with weak antiviral activity, which showed a chemical profile characterized by the presence of all eight types of oxygenated monoterpenes, and the highest amounts of sesquiterpenoids, both hydrocarbon and oxygenated sesquiterpenes. 

Cluster 2 grouped one EO (LoP) with strong antiviral activity and two EOs (LaCaf1 and PaS1) with weak activity, which showed higher amounts of monoterpene hydrocarbons and lower amounts of oxygenated monoterpenes and sesquiterpenes, compared to cluster 1. 

Cluster 3 grouped the three EOs of *L. origanoides* (LoTC, LoTf and LoC) with weak antiviral activity, which showed higher amounts of phenolic monoterpenes, lower amounts of sesquiterpene hydrocarbons, and had no five oxygenated monoterpenes (aldehydes, ketones, ethers, esters, and peroxides), compared to cluster 1. 

A one-way ANOVA of the DENV-2 CPE values of EOs in each cluster revealed significant differences between clusters 1 and 3 (Figure 4C, One-way ANOVA: F_2,9_ = 6.6092, *p* = 0.023, post hoc Tukey-Kramer test *p* = 0.026) but not clusters 1 and 2, suggesting a relation between higher content of non-phenolic oxygenated monoterpenes and sesquiterpenes, and higher antiviral activity.

### 2.4. Molecular Interactions between EO Compounds and DENV-2 Proteins

The DENV particle contains an RNA genome and capsid surrounded by a lipid envelope, which contains the envelope (E) and pre-membrane (prM/M) proteins [30]. Copies of the capside protein (C) form the viral capsid. Docking analysis was performed to predict the binding affinities between 68 sesquiterpenes and 53 monoterpenes, identified in the fourteen EOs, and the E, C, and prM/M proteins of DENV-2. A binding energy below the upper threshold of −6.8 kcal/mol was considered a cutoff value for predicting the binding affinity between ligand and target [31]. Appendix A presents the AutoDock Vina binding energies values of EOs terpenes with DENV-2 proteins 

The DENV-2 E protein is organized into three structural domains. The beta-octylglucoside detergent binding site (the "βOG pocket") is in the hinge region formed by DI and DII domains. It plays an essential role in viral entry by mediating fusion between virus and host cell membranes [30,32]. The docking analyses predicted the binding affinities between 38 terpenes and the DENV-2 E protein, which have formed hydrophobic bonds with amino acid residues (especially Thr48, Tyr137, Leu191, Phe193, and Phe279) of the βOG pocket.

Twenty sesquiterpene hydrocarbons docked to DENV-2 E, of which thirteen were bicyclic (−8.73 to −6.91 kcal/mol), five were tricyclic (−8.13 to −7.62 kcal/mol), and two were monocyclic compounds (−7.83 to −6.91 kcal/mol). Two bicyclic sesquiterpene alcohols docked to DENV E (−7.07 to −6.91 kcal/mol), vetiselinenol had formed a hydrogen bond with Thr280 (4.6 Å). Fifteen monoterpenes were predicted to dock into the E-βOG pocket, all of them were monocyclic compounds, of which eight were oxygenated (−7.52 to −6.98 kcal/mol) and seven were hydrocarbons (−7.60 to −6.99 kcal/mol). Three monoterpenes had formed hydrogen bonds: carvone with Thr48 (4.3 Å); carvacrol with His282 (4.6 Å); *trans*-dihydrocarvone with His27 (4.8 Å) and Thr48 (4.18 Å). Figure 5A,B show the predicted binding site of representative terpenes to DENV-2 E protein.

Based on our docking analysis, the potential monoterpenes and sesquiterpenes candidates that may inhibit DENV-2 E are listed in Table 4. The top five sesquiterpenes were *cis*-calamenene, δ-cadinene, α-cadinene, α-guaiene, and γ-cadinene, and the top five monoterpenes were α-phellandrene, carvacrol, carvone, γ-terpinene, and limonene.

The DENV C protein is organized into two pairs of antiparallel helical interfaces namely, α2–α2′ and α4–α4′, and amino acid residues of the α1 and α3 helices form a concave hydrophobic cleft [30]. The docking analyses predicted binding affinity between α-phellandrene (−7.60 kcal/moL) and *cis*-calamenene (−6.91 kcal/mol) with DENV-2 C. α-Phellandrene had formed hydrophobic interactions with amino acid residues (especially Ala49, Ala52, Phe53, Phe56, Phe84) of the α2-α2′ interface. *Cis*-Calamenene had formed hydrophobic interactions with amino acid residues (especially Leu44, Phe47, and Met48) of the loop that connects α1 and α3 helices (Figure 5C,D). 

The DENV prM/M protein is believed to be a chaperon for the E protein to assure its conformational changes to produce infectious viral particles [30]. The docking analysis did not predict terpenes to dock the prM/M protein of DENV-2 (docking score: −5.55 to −4.12 kcal/mol).

## 3. Discussion

Among the medicinal plants growing in Colombia, we have selected six species used in both, folk medicine, and the food and pharmaceutical industries [19,20,21,22,23,24]. We have previously studied the chemical compositions of EOs distilled from *L. alba* and *L. origanoides* [33,34], and the chemical profile of the EO samples analyzed in this study resembled, with minor differences. Generally, *L. alba* and *L. origanoides* EOs are classified as monoterpene-rich EOs. The two EO samples of *T. diffusa* in this study showed differences in their chemical constituents from each other and from another sample from Colombia that we analyzed in a previous study [35]. Chemical variability among *T. diffusa* EOs due to geographic location, vegetative state of the plant, and storage time of the harvested is documented [21]. The chemical composition of the EOs of *P. aduncum, O. basilicum*, and *V. curassavica* from Colombia is not yet documented. Three piperitone-rich chemotypes of *P. aduncum* have been proposed [36,37], and the EO sample analyzed in this study could be classified as piperitone-rich oil. The EO of *O. basilicum* in this study could belong to both the linalool-rich and European chemotypes [38], characterized by high amounts of linalool. The EO of *V. curassavica* in this study is similar in chemical composition to EO samples from Brazil [39]. 

The results of this study showed variability in antiviral activity against DENV among EOs from the same and different plant species. The EOs of *L. alba* (citral and carvone chemotypes), *L. origanoides* (phellandrene chemotype), and *T. diffusa* showed strong antiviral activity against DENV-1 and DENV-2. These EOs reduced the CPE of both virus serotypes at IC_50_ < 100 µg/mL and SI > 5.0, which are parameters for EOs with antiviral efficacy in vitro [12,13,40]. In a previous study, *L. alba* EO also exhibited an antiviral effect against all four DENV serotypes in a plaque reduction assay [41]. We are unaware of reports on the antiviral activity of *L. origanoides* EO against DENV. We have previously documented the antiviral activity of this EO against the yellow fever virus [42], a pathogenic virus similar in structure to DENV. The EO of *T. diffusa* (2016) showed the highest antiviral activity (SI of 14.3), to our knowledge, there are no published studies demonstrating the antiviral activity of this EO against human viruses. *O. basilicum* EO showed weak anti-DENV activity; in vitro antiviral efficacy of this EO against pathogenic enveloped viruses has been documented [43,44,45]. *P. aduncum* EO showed weak anti-DENV activity; we have not found studies on the antiviral efficacy of this EO on enveloped viruses. 

It has been well demonstrated that the chemistry of EO determines the bioactivity of EOs [9,10]. In this study, the combination of data from the antiviral CPE reduction assay and GC/MS analysis suggests that terpene content can influence the antiviral activity of the EOs against DENV. The anti-DENV activity of monoterpene-rich EOs increased in oils with high concentrations of monoterpenes alcohols, ethers and ketones, and sesquiterpene hydrocarbons. Conversely, the anti-DENV activity decreased in EOs with higher phenolic monoterpene concentrations and lower amounts of sesquiterpenes and monoterpene alcohols. The antiviral activity of sesquiterpene-rich EOs increased with the concentrations of oxygenated monoterpenes. As for *T. diffusa* EO samples, TdS2 showed strong anti-DENV activity, whereas TdS1 was found to be inactive. TdS2 but not TdS1 contains four oxygenated monoterpenes (terpinen-4-ol, α-terpineol, carvacrol, and thymol), one oxygenated sesquiterpene (zierone), and two sesquiterpene hydrocarbons (δ-cadinene and α-humulene), of which six compounds were predicted to bind to the E and C proteins of DENV. The different chemical compositions could partly explain the higher antiviral activity of TdS2 EO. *V. curassavica* EO was inactive, whereas *O. basilicum* EO showed weak antiviral activity; the former had a much lower content (1.7%) of oxygenated monoterpenes than the latter (69.4%). 

The type of monoterpenes and sesquiterpenes has been recognized as a factor influencing the antiviral activity of EOs against pathogenic enveloped viruses [12,13]. The cluster analysis of this study suggests that the oxygenated monoterpenes, especially alcohols, ketones, ethers, and sesquiterpene hydrocarbons, might be associated with the antiviral activity of the EOs against DENV. The DENV envelope layer mainly comprises lipid bilayers, and it has been well demonstrated that alcohol compounds can favorably interact with enveloped viruses and block their infectivity [46]. The sesquiterpene hydrocarbon fraction, rather than the oxygenated fraction, accounted for the antiviral action against the Herpes Simplex-2 virus, an enveloped human pathogenic virus [47]. Furthermore, a mixture containing monoterpene alcohols and sesquiterpene hydrocarbons derived from *Melaleuca alternifolia,* showed promising in vivo antiviral efficacy against West Nile virus, an enveloped virus similar in structure to DENV [48]. 

The mechanisms of action of EOs against viruses have not yet been fully elucidated. Data from addition time experiments suggest that EOs mainly inhibit virus adsorption on the cell by acting on cell-free viruses directly [12,13,14]. The antiviral assay in this study evaluated the ability of the EOs to interfere with virus adsorption to the cell. The reduction of DENV CPE suggests that EOs may have exerted their antiviral action by destroying the viral envelope or its masking and, consequently, blocking the interaction of viral particles with the cell membrane. We have found that EOs of *Lippia* spp. showed better antiviral action before than after viral adsorption [41,42]. EOs also have intracellular modes of action by blocking the viral particle formation and releasing them from the host cell [12,13,14]. EO of *L. alba* is rich in *trans-*β-caryophyllene and this isolated sesquiterpene reduced the Zika virus replication, in an enveloped virus similar in structure to DENV, by treatment after virus adsorption to the cell [49]. We could hypothesize that inactive or weakly active EOs in this study, might exhibit strong anti-DENV activity in a cell-based assay that examines the inhibitory effect after virus adsorption to cells. 

The in silico analysis of this study allows us to make observations about the contribution of terpene types to the antiviral action of the EOs against DENV. The analysis revealed a tendency of sesquiterpene hydrocarbons and oxygenated monoterpenes to bind to the E protein. The results support the hypothesis that nonpolar sesquiterpenes, rather than oxygenated sesquiterpenes, tend to interact more with viral proteins [13,50]. Sesquiterpenes and monoterpenes present in the EOs studied bound to the βOG pocket of the E protein, which is located at the region for the major conformational change during viral and cellular membrane fusion [29,51]. The βOG pocket has been established as a target for developing antivirals [51]. The efficacy of monoterpenes and sesquiterpenes to suppress DENV replication has not been systematically analyzed in cell-based assays. In a previous study [52], we demonstrated the intracellular antiviral action of β-caryophyllene, citral, *p*-cymene, and α-phellandrene to a greater extent than carvone, limonene, and nerol. Antiviral activities of numerous oxygenated monoterpenes and sesquiterpene hydrocarbons against many human pathogenic enveloped viruses have been well documented [11,12,13,16]. 

## 4. Materials and Methods

### 4.1. Plant Material and EO Distillation

All six plants used in this study were grown in the experimental plots at the Agroindustrial Pilot Complex of the National Center for Agroindustrialization of Aromatic and Medicinal Tropical Vegetal (CENIVAM). The taxonomic identification of these plants was performed at the Colombian National Herbarium, where their vouchers were placed. The plants were dried in the dark and leaves and stems were crushed and homogenized. The EOs were obtained by hydrodistillation (2 h) of plant material (80 kg) on a Clevenger apparatus as described elsewhere [30,32]. Fractional distillation of EOs under reduced pressure was carried out in a B/R Instruments (Easton, Easton, Maryland, USA) 800 High-Efficiency Micro Distillation device. The following auxiliary equipment was used: Edwards 8 vacuum pump (Edwards Vacuum, WS, Burgess Hill, UK), Alpha RA8 cooling bath (Lauda, Delran, NJ, USA), AREC-F magnetic stirring heating plate (VELP Científica, MB, Usmate Velate, Italy), and a magnetic stirring heating plate AREC-F (VELP Scientific, MB, Usmate Velate, Italy). EO (10–15 g) was deposited into the reboiler balloon of the micro-distiller, which was connected to the fractionation column. EO fractions were collected by opening the reflux valve each time the top temperature remained constant for 30 s. Reduced-pressure at 12 Torr (1.6 kPa) and 7 Torr (0.9 kPa) were used for *L. alba* citral and carvone chemotypes EO, respectively. As for *L. alba* carvone chemotype EO, the limonene-rich fraction (LaCaf1) was collected at 65–69 °C, and the carvone-rich fraction (LaCaf2) at 89–93 °C. For *L. alba* citral chemotype EO, the light fraction (LaCif) was collected at 115 °C. For the *L. origanoides* EO, the thymol-rich heavy fraction (LoTf) was collected at 10 Torr (1.3 kPa) and 65–69 °C. All EOs and their fractions were kept at 4 °C. For analysis of antiviral activity, stock solutions were prepared in dimethyl sulfoxide (DMSO, 0.5% final concentration) and stored at −20 °C until used.

### 4.2. Chromatographic Analysis

Essential oil analysis was performed by gas chromatography using mass spectrometric (GC/MS) and flame ionization detection (GC/FID) systems, as described elsewhere [33]. Before the GC analysis, each essential oil or its fraction was dried with sodium sulphate and weighed (50 mg), and then dissolved in dichloromethane (1 mL), with *n*-tetradecane (0.5 µL) as an internal standard. The injection volume was 2 µL in split mode (30:1) for GC/MS and GC/FID systems. For both systems, the injector temperature was maintained at 250 °C. A 6890 Plus Gas Chromatograph (Agilent Technologies, Palo Alto, CA, USA.) equipped with a mass selective detector MSD 5975 (Electron ionization, EI, 70 eV, AT, Palo Alto, CA, USA.), a 7863 automatic injector and an MSChemStation G1701DA data system (AT, Palo Alto, CA, USA.), were used. A fused-silica capillary column DB-5MS (J&W Scientific, Folsom, CA, USA.) of 60 m × 0.25 mm I.D., coated with 5%-phenyl poly (methyl siloxane) (0.25 μm-film thickness, d_f_) was used. Chromatographic conditions were as follows: GC oven temperature started from 45 °C (5 min) to 150 °C (2 min) at 4 °C/min, then to 250 °C (5 min) at 5 °C/min, and finally, to 275 °C (15 min) at 10 °C/min. The ionization chamber, the quadrupole, and the transfer line temperatures were set at 250 °C, 150 °C, and 285 °C, respectively. Helium (99.99%, AP gas, Messer, Bogota, Colombia) was used as carrier gas, with the initial inlet pressure of 113.5 kPa (27 cm/s linear velocity). Its volumetric flow rate was kept constant (1 mL/min). *n*-Alkane (C_8_–C_25_) and (C_8_–C_40_) mixtures (Sigma-Aldrich, San Luis, MO, USA) were used to obtain linear retention indices (LRI). Mass spectra and reconstructed ion chromatograms were obtained by automatic scanning of quadrupole radiofrequency in the *m*/*z* 30–300 mass range at 3.58 scan/s. Chromatographic peaks were checked for homogeneity with the aid of the mass spectra for the characteristic fragment ions. Identification of EO compounds was accomplished by comparison of their LRIs, measured on both non-polar (DB-5MS, 60 m  ×  0.25 mm  ×  0.25 μm) and polar (DB-WAX, 60 m  ×  0.25 mm  ×  0.25 μm) columns (J & W Scientific, Folsom, CA, USA), with those of the available standard compounds and by comparison of their mass spectral fragmentation patterns with those described in the scientific literature and of databases (Wiley-2008, NIST-2017, QUADLIB-2007). For the quantitative analysis (GC relative amounts, %), assuming that all compounds had the same response factor (Rf = 1), the EO samples, prepared as described above, were injected into the GC 6890 Plus Gas Chromatograph (AT, Palo Alto, CA, USA), coupled to the FID and the non-polar 5%-Ph-PDMS capillary column (J&W Scientific, Folsom, CA, USA) of the same dimensions (L, I.D., d_f_) as used for the GC/MS analysis. All chromatographic parameters for the GC/FID analysis (column, split ratio, injection volume, and temperatures) were the same as for the GC/MS experiments. The internal standard was used for result reproducibility checking (retention times and GC areas). 

### 4.3. Virus and Cells

Dengue virus type 1 (DENV-1) strain US/Hawaii/1944 and Dengue virus type-2 (DENV-2) New Guinea C strain (NGC) were used. Viruses were propagated in C6/36 *Aedes albopictus* cells (Pedro Kourí Institute for Tropical Medicine, La Habana, Cuba) and harvested on day 7 post-infection. Virus stock was titrated in BHK-21 cells (ATCC® CCL-10™) using the plaque method and stored at −80 °C until needed [38,39]. Vero cells (ATCC® CCL-81™) were cultured in Eagle’s Minimum Essential Medium (EMEM) containing 10% fetal bovine serum (FBS, Gibco, Grand Island, NY, USA) at 37 °C in the presence of 5% CO_2_. 

### 4.4. Cytotoxicity Assay

Cytotoxicity of EOs was evaluated in the crystal violet assay. Vero cells were seeded 96-well plates and allowed to adhere for 24 h at 37 °C and 5% CO_2_. Next, cells were treated with EO at seven-point concentrations (7.8 to 500 µg/mL) for 72 h at 37 °C. Not-treated cells and cells treated with dimethyl-sulfoxide were run in parallel as a negative and positive control, respectively. After washing with PBS, 100 µL of 0.05% crystal violet solution in 20% ethanol was added and cells were allowed to stain for 20 min at room temperature. After six washing with distilled water, the plates were aspirated and allowed to air-dry at room temperature, and 200 µL of methanol was added to each well for 20 min with shaking. The optical density at 570 nm at each well was measured on a microplate reader to quantify crystal violet staining. Each EO was analyzed in three independent assays in triplicate. 

### 4.5. Cytopathic Effect (CPE)-Based Antiviral Assay

A standard protocol was followed [29]. Vero cells were seeded 96-well plates and allowed to adhere for 24 h at 37 °C and 5% CO_2_. After washing with PBS, DENV (MOI of 1.0) in a culture medium containing EO at non-cytotoxic concentrations (3.12, 6.25, 12.5, 25, 50, and 100 µg/mL) was adsorbed on cells 1 h at 37 °C; 5% CO_2_. Next, the cells were washed with PBS, fresh media without EO was added, and the virus was allowed to replicate for 5 days at 37 °C under a 5% CO_2_ atmosphere. The next steps followed the cytotoxicity assay mentioned above for quantification of crystal violet staining to measure virus-induced CPE indirectly. Untreated DENV-infected cells and SDS-treated DENV-infected cells served as the negative and positive control, respectively. CPE of DENV was estimated using the following formula: DENV CPE (%) = [(OD_570_ of DENV-infected EO-treated cells/OD_570_ of non-infected non-treated cells) × 100]. Reduction of CPE was estimated as: DENV CPE reduction (%) = [(OD_570_ of DENV-infected EO-treated cells − OD_570_ of negative control)/(OD_570_ of positive control − OD_570_ of negative control) × 100]. The 50% inhibition concentration (IC_50_) was calculated by regression analysis (GraphPad Software, San Diego, CA, USA). Each EO was analyzed in three independent assays in triplicate. 

### 4.6. In Silico Analysis

Target proteins. The three-dimensional (3D) crystallographic structure of DENV-2 proteins was downloaded from the protein data bank (PDB ID 2FOM): E (ID: 1OAN), prM/M (ID: 3C5X), and C (ID: 1R6R). Three-dimensional protein structures were prepared using the PyMOL 2.3.0 software package (Schrodinger, Portland, OR, USA) by removing all the present water molecules and substructures. Energy minimization (1000 kJ/mol) was performed with the OPLS force field-AA/L implemented in the GROMACS 5.0 package (GROMACS development team, Groningen, NL) by using the steepest descent algorithm with a minimization step of 0.01 nm. Three-dimensional structures were prepared for molecular docking by adding hydrogen atoms, Kollman united atom type charges, and solvation parameters using AutoDock Tools [53]. Optimized structures were saved in PDB format and converted to PDBQT files with MGLTools (Center for computational structural biology, La Jolla, CA, USA).

Ligands preparation. All 125 EO compounds identified by GC/MS analysis were selected for docking analysis. Epigallocatechin gallate, with in silico and in vitro anti-DENV activities reported previously in the literature [54], was included as the positive control. Structures of ligands were retrieved from PubChem (https://pubchem.ncbi.nlm.nih.gov/ accessed on 30 August 2022) database. All ligands were prepared by adding hydrogen atoms and merging them with non-polar hydrogen atoms. Gasteiger partial charges were added, rotatable bonds were defined, and the energies were minimized using AutoDock Tools. Optimized structures were saved in MOL2 format and converted to PDBQT files with MGLTools. 

Docking analysis. Molecular docking was performed using the Autodock Vina 1.5.6 software [55]. Default parameters were used, and the search exhaustiveness parameter was set to 100. For each ligand, 27 docked conformations were generated using global docking simulations, i.e., the grid box was defined to cover all protein structures to search for the best binding site in the protein. Three simulations were performed for each ligand-protein pair by using seeds 6, 12, and 18. The binding free energy was approximated by the average of docking scores for each protein. Discovery Studio Visualizer v21.1.0.20298 (Dassault Systèmes, San Diego, CA, USA) was used to view ligand-protein interaction. The docking protocol was validated using a set of molecules (*n* = 8) reported in the literature as DENV-2 inhibitors interacting with the E protein (i.e., 2,4-disubstituted pyrimidine **2k**, 2,4-disubstituted pyrimidine **3a**, 2,4-disubstituted pyrimidine **5c**, castanospermine, celgosivir, NITD448, zosteric acid, and cinnamic acid). In addition, 3D structures were downloaded from the NCBI PubChem Database and were processed as mentioned. 

### 4.7. Correlation and Statistical Analysis 

To determine the relationship between chemical composition and antiviral activity, each EO was characterized under two independent parameters as follows; (i) chemical composition, the multivariate parameter with an absolute number (%) of different terpene types (Table 2); (ii) antiviral activity, DENV CPE as a unidimensional variable ranging from 0 to 100, low CPE values indicated strong antiviral activity. EOs with CPE ≤ 60 entered the analysis (*n* = 10). The samples were assigned into clusters according to their terpene compositions using an unsupervised self-organized Kohonen Map [56]. Each EO was represented by a vector with fifteen values represented by Equation (1) where *C* stands for its chemical composition formed by compounds *c_n_*. The algorithm was based on the distance *d* between vectors *C_i_* and a set of artificial neurons *N_j_* conforming to the network and was represented by vectors of the same size as the samples (Equation (2)). Using an iterative approach, the *N_j_* values were updated to minimize the distance *d* between neurons and samples (Equation (3)). As a result, EOs with similar terpene types will fall close to the same representative neuron, defining a given cluster. In contrast, EOs with dissimilar compositions will fall close to different neurons, indicating their membership in different clusters.
(1)$$C=\left[c_{1},c_{2},\ldots c_{15}\right]$$
(2)$$N=\left[n_{1},n_{2},\ldots n_{15}\right]$$
(3)$$d\left(C_{i},N_{j}\right)=\sqrt{{\left(c_{i1}-n_{j1}\right)}^{2}+{\left(c_{i2}-n_{j2}\right)}^{2}+\cdots {\left(c_{in}-n_{jn}\right)}^{2}}$$

A one-way ANOVA and a Tukey-Kramer post hoc test of CPE values were used to compare the antiviral potential of each resulting cluster, adopting a significance level of 0.0. Clustering and statistical analysis were performed using Matlab^®^ R2021b (Mathworks Inc., Natick, MA, USA).

## 5. Conclusions

This study reports for the first time differences in the antiviral activity of the EOs, against DENV, corresponding to their composition of monoterpenes and sesquiterpenes. Fourteen EO samples of six plant species grown in Colombia were studied, which were different in the content of terpenes. As for monoterpene-rich EOs, samples that had monoterpene alcohols (LaCi, citral chemotype) or ketones (LaCaf2, carvone-rich fraction) had the greatest antiviral activity among *L. alba* EOs; and the sample rich in both sesquiterpene hydrocarbons and oxygenated monoterpene (LoP, phellandrene chemotype) had the highest antiviral activity among *L. origanoides* EOs. As for sesquiterpene-rich EOs, samples that had monoterpene alcohols showed antiviral activity (EOs of *T, diffusa* 2016, *O. basilicum* and *P. aduncum*), whereas samples that had no monoterpene alcohols and very low amounts of other oxygenated monoterpene were inactive (*T. diffusa* 2019 and *V. curassavica* EOs). Sesquiterpene hydrocarbons and oxygenated monoterpenes present in the EOs with antiviral activity showed binding affinities with the E protein of DENV-2, which suggests that these terpene types could act as inhibitors of virus adsorption and entry into the cell. The results of this study provide information on the antiviral activity of EOs and their terpenes on DENV replication in vitro and could be used in the research and development of herbal medicines for the prophylactic treatment of severe dengue. Sesquiterpene hydrocarbons and oxygenated monoterpenes may serve as a starting material for the development of antiviral phytomedicines. Well-controlled activity-guided fractionation, mode-of-action elucidation studies, and pre-clinical trials are encouraged to establish the use of EO-based herbal medicines for preventing severe dengue. 

## Figures and Tables

**Figure 1 molecules-27-06844-f001:**
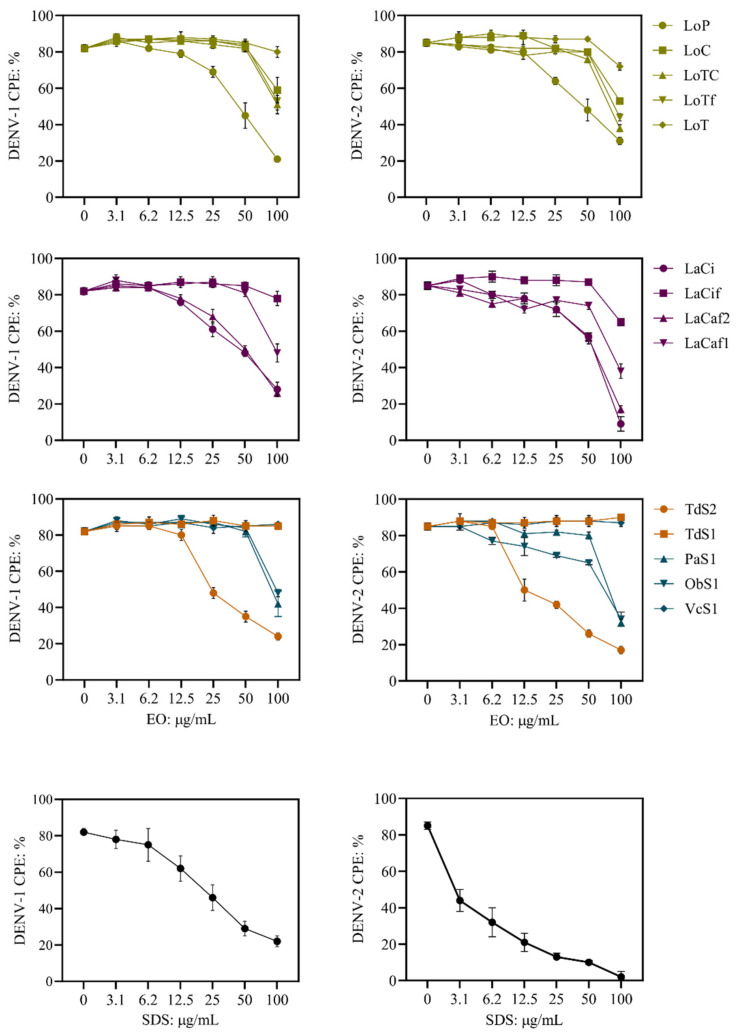
Results of cytopathic effect (CPE)-based assays to evaluate the antiviral activities of the essential oils against dengue viruses (DENV-1 and DENV-2). Quantification of crystal violet staining was performed to measure virus-induced CPE. % DENV-CPE = [(OD 570 of infected and treated cells/OD 570 of non-infected non-treated]. EO samples (Table 1): *L. origanoides* (LoP, LoC, LoTC, LoTf, and LoT); *L. alba* (LaCi, LaCif, LaCaf2, LaCaf1); *T. diffusa* (TdS1 and TdS2); O. *basilicum* (ObS1); *P. aduncum* (PaS1) and *V. curassavica* (VcS1). Sodium dodecyl sulfate (SDS) is an antiviral compound. Data are expressed as the mean ± SD from six independent measurements.

**Figure 2 molecules-27-06844-f002:**
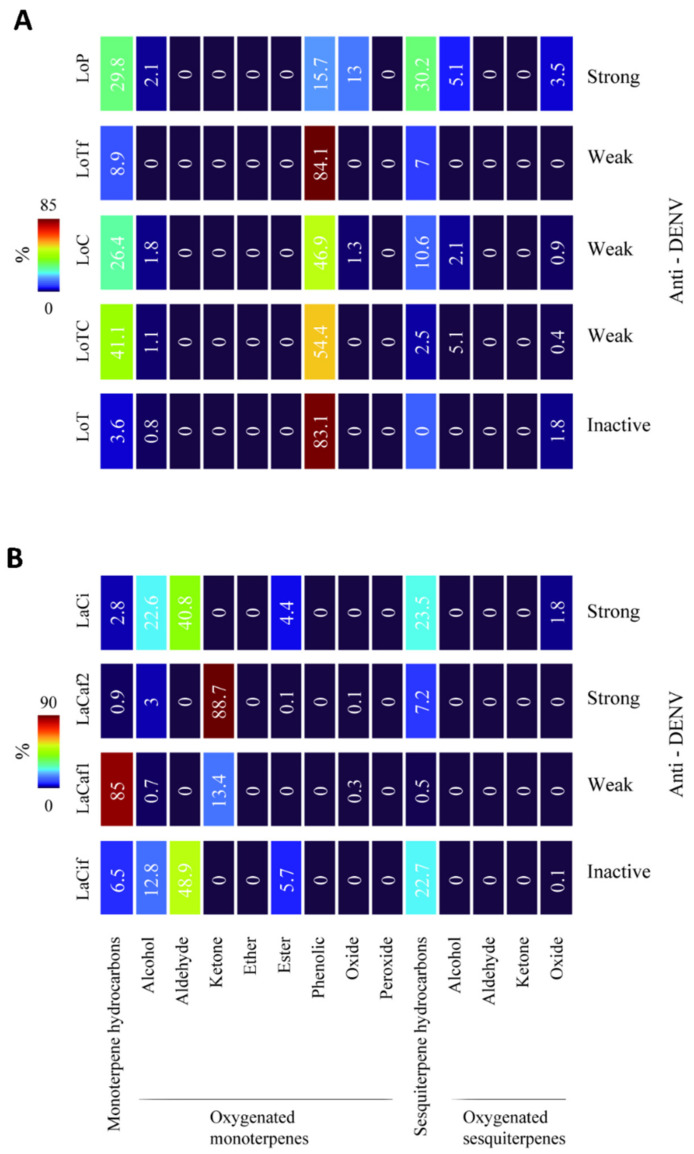
Antiviral activities of *Lippia* essential oils against dengue virus and terpene content. Heat maps representing the chemical composition in monoterpene and sesquiterpene types. Antiviral activity (Anti-DENV) was expressed as strong, weak, and inactive (Table 3). (**A**). EOs of *L. origanoides* chemotypes: neat EOs phellandrene EO (LoP); carvacrol (LoC); thymol-carvacrol (LoTC) and thymol (LoT); and thymol-rich fraction (LoTf). (**B**). EOs of *L. alba*: neat EO (LaCif) and light fraction (LaCif) of the citral chemotype; and limonene-rich fraction (LaCaf1) and carvone-rich fraction (LaCaf2) of the carvone chemotype.

**Figure 3 molecules-27-06844-f003:**
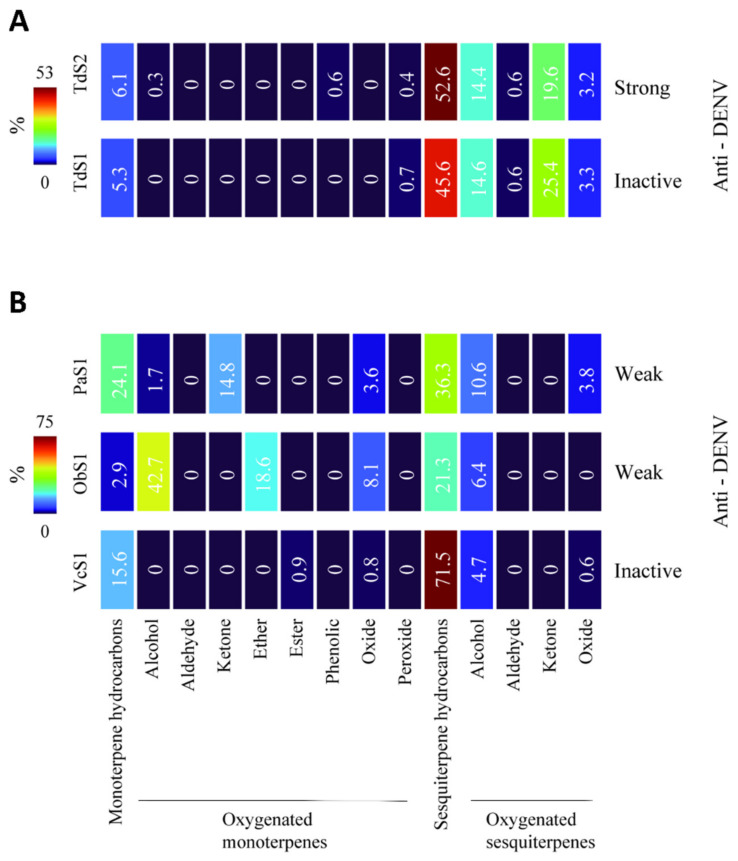
Antiviral activities of essential oils against dengue virus and terpene content. Heat maps representing the chemical composition in monoterpenes and sesquiterpenes types. Antiviral activity (Anti-DENV) was expressed as strong, weak, and inactive (Table 3). (**A**). Neat EOs of *T. diffusa* collected in 2019 (TdS1) and 2016 (TdS2). (**B**). Neat EOs of *O. basilicum* (ObS1) *P. aduncum* (*Pa*S1) and *V. curassavica* (VcS1).

**Figure 4 molecules-27-06844-f004:**
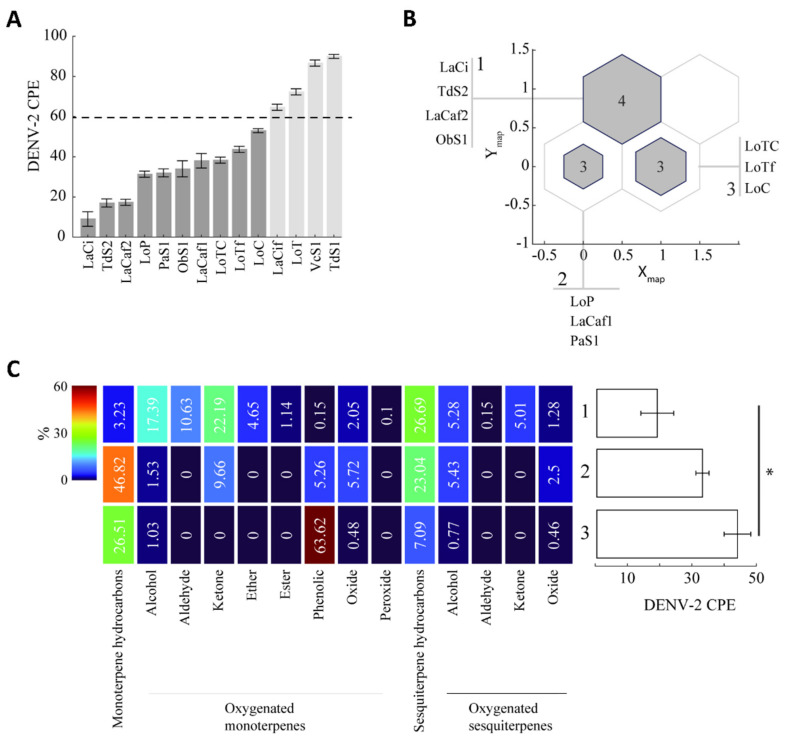
Clustering of essential oils based on monoterpene and sesquiterpene contents and their relation to antiviral activity. (**A**)**.** Cytopathic effect (CPE: %) of DENV-2 in Vero cells, EOs identifiers in Table 1. The dashed line indicates the CPE threshold for further clustering analysis, only EOs with CPE ≤ 60 (dark gray bars) were included. (**B**)**.** Clustering configuration of selected EOs in A using a 2 × 2 self-organized map. Three non-empty clusters were obtained. The number inside each cluster indicates the number of EOs it contains. (**C**). Left, heat map representing the chemical composition of clusters 1 (upper row), 2 (middle row), and 3 (lower row). Right, mean CPE of each cluster indicating statistical differences between clusters 1 and 3 (* One-way ANOVA: F_2,9_ = 6.6092, *p* = 0.023, post hoc Tukey-Kramer test *p* = 0.026).

**Figure 5 molecules-27-06844-f005:**
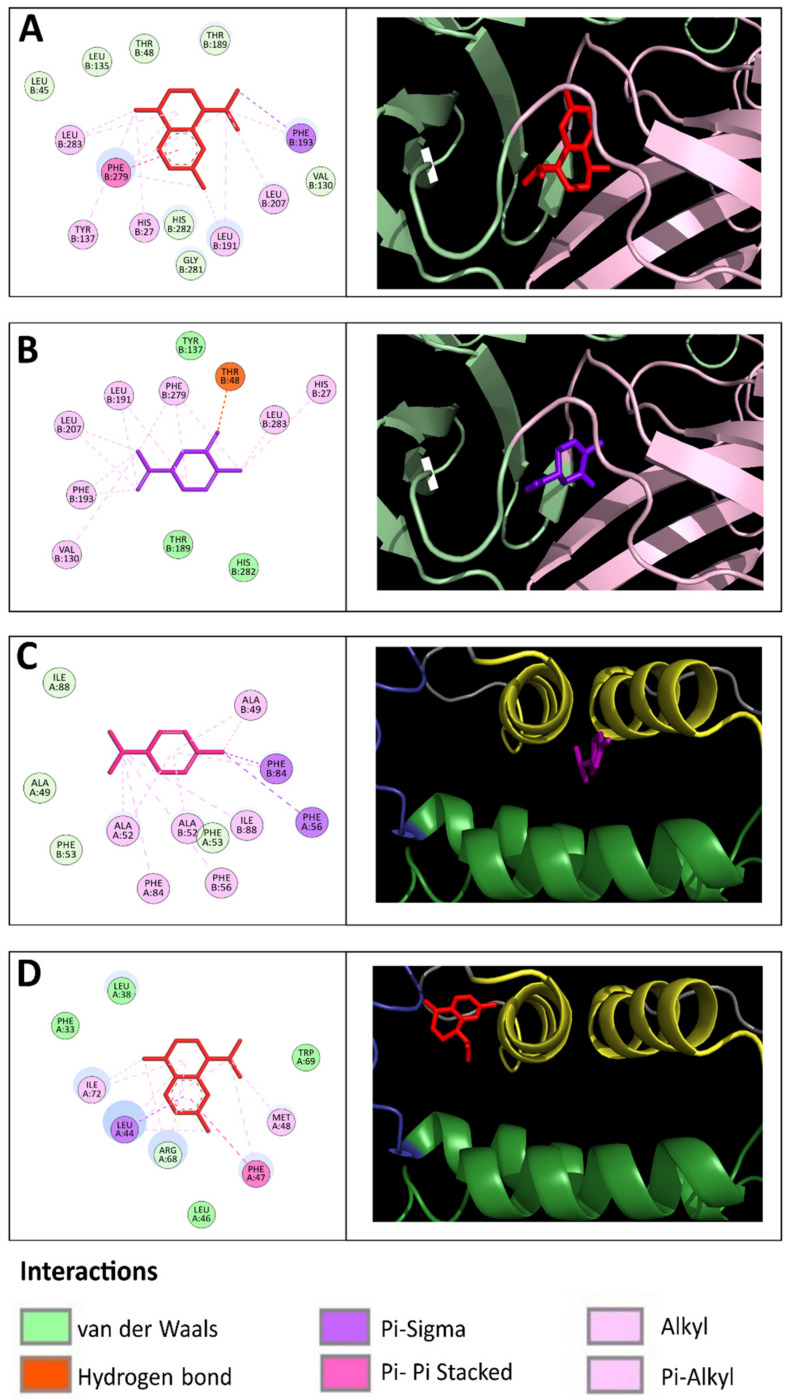
Lowest-energy docked poses of sesquiterpenes and monoterpenes with DENV-2 proteins. Docked *cis*-calamenene (**A**) and carvone (**B)** in complex with E protein showing bonds with amino acids at the hinge region. (**C**). Docked α-phellandrene in complex with C protein showing bonds with amino acids at the α2-α2′ dimer interface hinge region. (**D**). Docked *cis*-calamenene in complex with C protein showing bonds with amino acids at the loop that connect α1 and α3 helices.

**Table 1 molecules-27-06844-t001:** Essential oils studied in this work.

Plant Material	Voucher Number	EO Identifier	EO Characteristics *
*Lippia origanoides* Kunth	22035	LoP	Phellandrene chemotype, neat EO
	22034	LoC	Carvacrol chemotype, neat EO
	22039	LoTC	Thymol-carvacrol chemotype, neat EO
	22036	LoT	Thymol chemotype, neat EO
		LoTf	Thymol chemotype, thymol-rich fraction
*Lippia alba* (Mill.) N.E.Br. ex Britton & P. Wilson	22002	LaCi	Citral chemotype, neat EO
		LaCif	Citral chemotype, light fraction
	22031	LaCaf1	Carvone chemotype, limonene-rich fraction
		LaCaf2	Carvone chemotype, carvone-rich fraction
*Turnera diffusa* Willdenow	22032	TdS1	2019, neat EO
	22037	TdS2	2016, neat EO
*Piper aduncum* L.	22033	PaS1	Linalool chemotype, neat EO
*Ocimum basilicum* L.	22227	ObS1	Piperitone chemotype, neat EO
*Varronia curassavica* Jacq.	20892	VcS1	Neat EO

***** Based on the chemical profile analysis and the extraction method (see Materials and Methods).

**Table 2 molecules-27-06844-t002:** The main chemical constituents of essential oils or their fractions studied in this work.

Compound	LRIs (DB-5MS Column)		LaCi	LaCif	LaCaf1	LaCaf2	LoP	LoC	LoT	LoTf	LoTC	TdS1	TdS2	PaS1	ObS1	VcS1
	Exp.	Lit.														
α-Pinene *	935	932 ^a^	0.3	0.2	0.2	-	1.7	0.4	-	-	0.4	-	-	4.6	0.5	9.4
α-Phellandrene	1005	1002 ^a^	0.1	-	-	-	7.1	-	-	-	0.5	-	-	4.4	-	-
*p*-Cymene *	1027	1024 ^a^	-	-	-	-	12.6	14.4	2.3	2.0	19.1	3.0	3.6	3.0	-	-
Limonene *	1034	1029 ^a^	2.4	5.9	82.2	0.9	2.1	0.3	-	-	0.9	-	-	6.0	-	0.8
1,8-Cineol *	1036	1031 ^a^	-	-	-	-	13.0	1.3	-	-	-	-	-	3.6	8.1	0.8
γ-Terpinene	1061	1059 ^a^	-	-	-	-	2.4	5.3	0.9	6.9	9.2	0.6	0.7	0.8	-	-
Linalool *	1099	1096 ^a^	1.1	2.5	0.4	0.5	0.7	1.0	-	-	0.3	-	-	0.4	42.7	-
Estragole *	1203	1196 ^a^	-	-	-	-	-	-	-	-	-	-	-	-	18.6	-
Neral *	1246	1252 ^b^	11.9	18.1	-	-	-	-	-	-	-	-	-	-	-	-
Carvone *	1259	1258 ^c^	-	-	12.2	78.2	-	-	-	-	-	-	-	-	-	-
Geraniol *	1260	1240 ^b^	19	8.1	-	-	-	-	-	-	-	-	-	-	-	-
Piperitone	1265	1264 ^c^	-	-	-	4.8	-	-	-	-	-	-	-	14.8	-	-
Geranial *	1272	1270 ^b^	24.5	24.8	-	-	-	-	-	-	-	-	-	-	-	-
Thymol *	1290	1290 ^a^	-	-	-	-	14.0	8.0	75.3	82.9	49.4	-	0.2	-	-	-
Carvacrol *	1300	1298 ^a^	-	-	-	-	0.9	35	4.9	1.2	2.7	-	0.4	-	-	-
Piperitenone	1347	1343 ^a^	-	-	0.3	0.3	-	-	-	-	-	-	-	14.8	-	-
α-Copaene	1385	1376 ^a^	-	-	-	0.3	0.6	0.7	-	-	-	-	-	2.9	-	7.0
*trans*-β-Caryophyllene *	1433	1427 ^c^	9.1	13.3	-	0.1	15.1	4.4	5.4	7.0	1.6	4.0	4.9	7.4	0.9	19.2
α-Humulene *	1467	1468 ^c^	2.8	2.8	-	-	8.1	1.1	3.2	-	0.9	-	0.4	1.5	2.5	2.7
Aristolochene	1483	1488 ^a^	-	-	-	-	-	-	-	-	-	17.9	20.9	-	-	-
Germacrene D	1492	1481 ^a^	4.3	1.5	0.1	-	0.9	-	-	-	-	-	-	1.7	4.9	12.3
β-Selinene	1502	1490 ^a^	-	-	-	-	0.5	0.3	-	-	-	5.2	5.8	-	-	-
Valencene	1503	1496 ^a^	-	-	-	-	-	-	-	-	-	7.4	6.5	1.2	-	-
*trans*-β-Guaiene	1517	1502 ^a^	2.2	-	-	-	-	-	-	-	-	-	-	-	-	11.8
Dehydrofukinone	1827	1820 ^c^	-	-	-	-	-	-	-	-	-	25.4	19.3	-	-	-

LRIs, linear retention indices calculated using *n*-alkanes C8–C25 mixture on the DB-5MS (non-polar) column. *L. alba*: neat EO (LaCif) and light fraction (LaCif) of citral chemotype; and limonene-rich fraction (LaCaf1) and carvone-rich fraction (LaCaf2) of carvone chemotype. *L. origanoides*: neat EO of phellandrene chemotype EO (LoP), carvacrol (LoC), thymol-carvacrol (LoTC), and thymol (LoT) chemotypes; and thymol-rich fraction (LoTf) of the thymol chemotype. Neat EOs of *T. diffusa* collected in 2019 (TdS1) and in 2016 (TdS2). Neat EOs of *O. basilicum* (ObS1), *P. aduncum* (PaS1) and *V. curassavica* (VcS1). Exp. Experimental. Lit. Literature ^a^ Adams, 2007 [26]; ^b^ Babushok et al., 2011 [27]; ^c^ NIST 2017 [28]. * Standard compounds were used for confirmatory identification.

**Table 3 molecules-27-06844-t003:** The activity of the essential oils tested against dengue virus serotypes in the cytopathic effect (CPE) assay.

Plant Material	Essential Oil Identifier	DENV-1	DENV-2
*Lippia origanoides*	LoP	Strong: IC_50_ (SI): 77 ± 1.1 (6.6)	Strong: IC_50_ (SI): 75 ± 1.0 (6.8)
	LoC	Weak	Weak
	LoTC	Weak	Weak
	LoTf	Weak	Weak
	LoT	Inactive	Inactive
*Lippia alba*	LaCi	Strong: IC_50_ (SI): 78 ± 1.1 (5.5)	Strong: IC_50_ (SI): 67 ± 1.2 (6.4)
	LaCaf2	Strong: IC_50_ (SI): 82 ± 1.1 (5.8)	Strong: IC_50_ (SI): 72 ± 1.1 (6.6)
	LaCaf1	Weak	Weak
	LaCif	Inactive	Inactive
*Turnera diffusa*	TdS2	Strong: IC_50_ (SI): 54 ± 1.1 (7.7)	Strong: IC_50_ (SI): 29 ± 1.1 (14.3)
	TdS1	Inactive	Inactive
*Piper aduncum*	PaS1	Weak	Weak
*Ocimum basilucum*	ObS1	Weak	Weak
*Verronia curassavica*	VcS1	Inactive	Inactive

**Table 4 molecules-27-06844-t004:** Essential oil sesquiterpenes and monoterpenes with the lowest binding free energy for DENV-2 E protein.

N^o^	Compound	Kcal/mol	StructuralFormula	Amino Acid Residues. H-bond in Bold Font
**1**	*cis*-Calamenene	−8.73	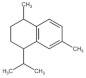	Thr189, Leu19, Phe193, Leu167, Phe279.
**2**	δ-Cadinene	−8.41	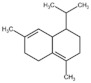	Leu45, Thr48, Leu135, Tyr137, Thr189, Leu191, Phe193, Phe279, Leu283.
**3**	α-Cadinene	−8.28	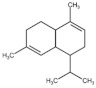	Leu45, Thr48, Leu135, Tyr137, Leu191, Phe193, Phe279, Leu283.
**4**	α-Guaiene	−8.26	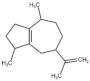	Glu26, Leu45, Thr48, Tyr137, Leu191, Phe193, Phe279, Leu283.
**5**	γ-Cadinene	−8.19	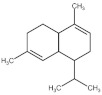	Leu45, Thr48, Leu135, Tyr137, Leu191, Phe193, Ohe279, Leu283.
**6**	Viridiflorene	−8.13	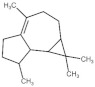	Thr48, Leu135, Tyr137, Thr189, Leu191, Phe193, Leu207, Phe279, Leu283.
**7**	α-Selinene	−7.98	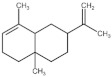	Thr48, Leu135, Thr189, Leu191, Phe193, Phe193, Phe279, Leu283.
**8**	δ-Amorphene	−7.96	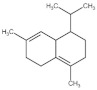	Leu45. Thr48, Leu135, Tyr137, Leu191, Phe193, Phe279, Leu283.
**9**	β-Bourbonene	−7.95	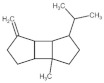	Thr48, Val130, Leu135, Tyr137, Leu191, Phe193, Leu207, Phe279, Leu283.
**10**	α-Gurjunene	−7.83	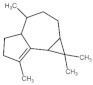	Thr48, Val130, Leu135, Tyr137, Thr189, Leu191, Phe193, Leu207, Phe279.
**11**	α-Phellandrene	−7.60	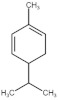	Thr48, Leu135, Tyr137, Thr189, Phe193, Phe279, Leu283
**12**	Carvacrol	−7.31	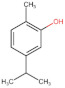	Thr189, Leu191, Leu207, Phe279, Leu283, His282
**13**	Carvone	−7.29	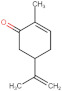	Thr48, Thr189, Leu191, Phe193, Leu207, Phe279.
**14**	γ-Terpinene	−7.28	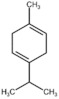	Thr48, Val130, Leu191, Leu207, Phe279, Leu283.
**15**	*p*-Cymene	−7.27	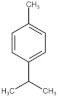	Thr48, Val130, Leu191, Leu207, Phe279, Leu283.
**16**	Limonene	−7.24	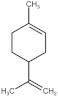	Thr48, Leu135, Phe193, Phe279, Leu283.
**17**	*trans*-Dihydrocarvone	−7.23	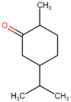	His27, Thr48, Leu191, Phe193, Leu207, Phe279.
**18**	Thymol-methyl-ether	−7.11	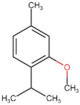	Leu191, Phe193, Leu207, Phe279.
**19**	α-Terpinene	−7.10	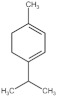	Thr48, Val130, Leu191, Phe279, Leu283.
**20**	Terpinolene	−7.03	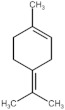	Thr48, Leu135, Thr189, Phe193, Leu207, Leu277, Phe279.

## Data Availability

Not applicable.

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
