# Peer review of "Essential Oils from Colombian Plants: Antiviral Potential against Dengue Virus Based on Chemical Composition, In Vitro and In Silico Analyses"

_molecules, 2022, doi:10.3390/molecules27206844_

Round 1

Reviewer 1 Report

Please see the file

Author Response

Manuscript ID: molecules-1919658

Authors: Lina Silva-Trujillo, Elizabeth Quintero-Rueda, Elena E Stashenko, Sergio Conde-Ocazionez, Paola Rondón-Villarreal, Raquel E Ocazionez *

Responses to Reviewer 1

 We thank the reviewer for his/her valuable suggestions and for recognizing the value of the presented data. We incorporated all the suggestions and we are confident of having improved the quality of our manuscript. We present here our replay to each of comment separately.

Comment 1. Line 36.

“..Please chang by “developed”..”

 We have rewritten the phrase using developing and underdeveloped countries

Comment 2. Line 61.

“..Accordinf the plant list site Turnera diffusa belong to Passifloraceae..”.

 The genus Turnera has been assigned to the families Turneraceae and Passifloraceae (Mabberley, D.J., 2008. Mabberley's Plant-Book, third ed. Cambridge University Press, Cambridge. Thulin, M. et al. 2012. Taxon 61,308–323). However, we agree and follow the reviewer's suggestion. We have changed Turneraceae to Passifloraceae.

Comment 3. Line 92.

“.The geraniol had the highest amount from the nonanal, so the main compounds are geranial and geraniol and not geranial and nerale.”.

We have replaced neral with gernaiol

Comment 4. Line 98.

“ According Table S1 piperitone was only present in LaCaf2 with an amount of 4.8%..”

 Our GC-MSS analysis identified piperitone (14.8%) as the main constituent of the essential oil (PaS1) distilled from Pipper auduncum. The composition of PaS1 is similar to that of P. auduncum samples from Latin American countries (Rodriguez, E.J. et al. Nat. Prod. Commun. 2013, 8, 1325-1328. da Silva JK., et al., Int. J. Mol. Sci. 2017, 18, 2571). We made a mistake in writing the chemical composition presented in Table S1. We have include the value 14.8% in Table S1.

Comment 5. Line 127. in a dose-dependent manner..

“.Add these information in the table or in supplementary material..”

 We have included “in a dose-dependent manner” in the Table 3 legend

Comment 6. Lines 132-133….”…(DENV CPE: 32 to 53%, respect 132 to 100% untreated control).

“Aslo here, these result are omitted in table, so please add them”

 We have added “..DENV CPE: 32 to 53%, respect 132 to 100% untreated control)..” in the Table 3 legend .

Comment 7. Comparisons of antiviral activities against DENV-2 and terpene content (lines 151-160).

(….2.5 to 10.6%). “This value it is the percentage of SH in which sample? The same for 10.6%...”

(…0.8 to 1.8%). Add in which samples this values were found...”

(….46.9 to 84.1%). “..do the same as previous...”

( ...8.9 to 41.1%). “..do the same as previous...”

(…14.1%) “…do the same as previous..”

(...... 22.6%). where you found this percentage.”

(…. 1.8%). “..where you found this percentage”

(.. 45.6%). “..where you found this percentage

(…69.4% vs 0.8%).“..where you found this percentage”

 The percentages of the terpenes types in each EO sample analyzed were presented in Table 2, first version of the manuscript. In the second version of the manuscript, the percentages are shown in heat maps in Figure 2 and 3, because the Table 2 was modified according to a reviewer's comment.

  • The percentage 6 corresponds to geraniol (19%), nerol (2.5%) and linalool (1.1%) present in the LaCi (L. alba citral chemotype) sample (Table 2, first version of the manuscript; Figure 2A, second version of the manuscript).
  • The percentage 1.8 corresponds to the content of sesquiterpene oxides in LaCi ( alba citral chemotype) sample (Table 2, first version of the manuscript; Figure 2A, second version of the manuscript).
  • The percentage 6 corresponds to the content of sesquiterpene hydrocarbons in the TdS1 (T. diffusa) sample (Table 2, first version of the manuscript; Figure 3A, second version of the manuscript).
  • The percentage 69.4 corresponds to alcohol (42.7%), ethers (18.6%) and oxides (8.1%) monoterpenes present in the ObS1 ( basilicum) sample (Table 2, first version of the manuscript; Figure 3B, second version of the manuscript).
  • The percentage 0.8 corresponds to the content of monoterpene oxides in VcS1 ( curassavica) sample (Table 2, first version of the manuscript; Figure 3B, second version of the manuscript).

For clarity, we have rewritten the text making comparisons of essential oil antiviral activities and terpene content (lines 151-160, first version of the manuscript).

Reviewer 2 Report

The work under review presents preliminary evaluation of six medicinal plants grown in Colombia as antiviral agents, tested in vitro, and evaluated in silico. If the manuscript is considered for publication, there are some points to be taken into consideration.

1.       Title: It is too long. Consider changing it into a more concise version. A suggestion might be “Antiviral evaluation of the essential oils from six Colombian medicinal plants”.

2.       Introduction: At some points, especially in the first two paragraphs, it seems vague. The required literature is cited; however, it is not clearly mentioned why that material was selected for example and why it is important, or worth of investigation. The link is somehow missing. Nonetheless, the length of the text should be kept as it is, if possible, but to the point.

3.       Experimental:

i.                     Table 1: the samples characterized as “enriched” in a component, are not really enriched but just “rich”, as obtained by fractional distillation, according to the authors. Why use fractional distillation for some samples in the first place? Explain clearly if according to a preliminary in silico study, a volatile fraction, abundant in some specific component, was predicted to be more potent for instance.

ii.                   The quantification of the constituents was calculated by GC-MS? Without internal/external standards whatsoever? Analysis of the essential oils by GC-FID is necessary.

iii.                 Why were the essential oils diluted in tetradecane and DCM prior to GC-MS analysis and not in a nonpolar solvent such as n-pentane or n-hexane?

iv.                 No additional real time PCR for the estimation of the virus yield? The CPE showed promising preliminary results for some essential oils; however, the most active samples could have been tested again to verify the results by rRT-PCR. Some pure constituents should have been tested as well, either because they were indicated by the in silico study or as major compounds of a relatively active essential oil.

4.       Results:

i.                     Figure 1: It could be included in the supplementary material.

ii.                   Figures 2 and 3: They show the antiviral effect on one virus serotype. Why are the results from the second serotype not included in the figure, they should show some variations.

iii.                 Table 2: The presentation of the grouped components in a separate table is redundant at that point, since the following heatmaps show clearly the distribution of the chemical groups in each sample. Instead, a table including the major compounds in order of elution would be more suitable.

iv. The reference antiviral agent should be included in the tables and the charts for comparison.

5.       Supplemental material:

The table presenting the chemical constitution of the essential oils seems erratic. The compounds are not presented in their elution order from a DB-5MS column as stated by the authors, however their retention indices are calculated on a DB-5MS column and it is confusing. Their order in the table should be changed according to their order of elution.

6.  The text requires extensive linquistic improvement and needs to be checked for some typos also, such as  line 451: seeded in, Table 3. basilicum.

Author Response

Manuscript ID: molecules-1919658

Authors: Lina Silva-Trujillo, Elizabeth Quintero-Rueda, Elena E Stashenko, Sergio Conde-Ocazionez, Paola Rondón-Villarreal, Raquel E Ocazionez *

RESPONSES TO REVIEWER 2

 We thank the reviewer for his/her valuable suggestions and for recognizing the value of the presented data. We incorporated all the suggestions and we are confident of having improved the quality of our manuscript. We present here our replay to each of comment separately.

Comment 1. Title

“It is too long. Consider changing it into a more concise version. A suggestion might be “Antiviral evaluation of the essential oils from six Colombian medicinal plants”.

 We have rephrased the title as:

Essential oils from Colombian plants: antiviral potential against dengue virus based on chemical composition, in vitro and in silico analyses

Comment 2. Introduction.

At some points, especially in the first two paragraphs, it seems vague. The required literature is cited; however, it is not clearly mentioned why that material was selected for example and why it is important, or worth of investigation. The link is some how missing. Nonetheless, the length of the text should be kept as it is, if possible, but to the point.”

 We have rephrased the text of introduction, mentioning why that material was selected and why is worth of investigation. We have included a reference (number 3, second version of the manuscript).

Comment 3. Results

“Table 1: the samples characterized as “enriched” in a component, are not really enriched but just “rich”, as obtained by fractional distillation, according to the authors. Why use fractional distillation for some samples in the first place?”.

We have replaced enrich with rich in Table 1, figure legends, and text

We analyzed neat and fractionated samples of three Lippia sp. essential oils. We was interested in the variation of the antiviral activity due to increase in limonene, carvone, and thymol, which are main constituents of Lippia alba and Lippia origanoides essential oils.

Comment 5. Methods.

The quantification of the constituents was calculated by GC-MS?. Without internal/external standards whatsoever?, Analysis of the essential oils by GC-FID is necessary”.

We have rewritten section "4.2 GC chromatographic analysis" (Methods) to describe in more detail the quantification of the EC components. GC-FID analyses were carried out.

Comment 6. Methods.

Why were the essential oils diluted in tetradecane and DCM prior to GC-MS analysis and not in a nonpolar solvent such as n-pentane or n-hexane?...”

 N-tetradecane and dichloromethane (DCM) were used as internal standar a solvent, respectively, as described in section "4.2 GC chromatographic analysis".

Comment 7. Results.

“No additional real time PCR for the estimation of the virus yield? The CPE showed promising preliminary results for some essential oils; however, the most active samples could have been tested again to verify the results by rRT-PCR. Some pure constituents should have been tested as well, either because they were indicated by the in silico study or as major compounds of a relatively active essential oil”.

The CPE-based assay in Vero cells is a well-recognized assay that has been used in antiviral drug discovery against dengue viruses and several pathogenic viruses (Smee D., et al., 2017. http://dx.doi.org/10.1016/j.jviromet.2017.03.012; Yan K, et al. 2020. doi: 10.1186/s12985- 021-01587-z; Sing S., et al. 2015. doi:10.1016/j.bmcl.2015.02.059 ). CPE refers to the adverse effect on cultured cells associated with virus multiplication. It is plausible that essential oils are capable of inducing cellular stress and thereby inhibiting virus replication in a non-specific manner. However, the concentrations of essential oil that caused inhibition of CPE did not affect the viability of cells cultured for 3 days in the essential oil-containing medium. Furthermore, the cells were exposed to the essential oil for only 1.5 h in the antiviral assay. Overall, the results of the antiviral potential of the tested essential oils measured in the CPE assay can be reliable. Correlation between the antiviral effect measured in the CPE assay and the plaque reduction assay (for estimation of virus yield) is documented (Schmidtke et al. 2001. doi: 10.1016/s0166-0934(01)00305-6).

The CPE-based assay in Vero cells is a well-known assay that has been used in antiviral drug discovery against dengue virus and several pathogenic viruses (Smee D., et al., 2017. http://dx.doi.org/10.1016/j.jviromet.2017.03.012; Yan K, et al. 2020. doi: 10.1186/s12985- 021-01587-z; Sing S., et al. 2015. doi:10.1016/j.bmcl.2015.02.059). It is plausible that an essential oil is capable of inducing cellular stress and thereby inhibiting virus replication in a non-specific manner. However, the concentrations of the tested essential oil that caused inhibition of DENV-CPE did not affect the viability of the treated cells for 72 h in the cytotoxic assay. Furthermore, the cells were exposed to the essential oil for only 1.5 h in the antiviral assay. Overall, the results of the antiviral potential of the tested essential oils measured in the CPE assay can be reliable. Correlation between the results of the CPE assay and the plaque reduction assay, which measures the performance of the virus, is documented. (Schmidtke et al. 2001. doi: 10.1016/s0166-0934(01)00305-6).

Our study focused on the influence of the chemical composition of essential oils on the variation of their antiviral action against dengue virus, rather than on the identification of the most active essential oils and terpenes. Our study reports for the first time findings that may contribute to the understanding of this influence. We agree with the reviewer that further analyses are needed to conclude more precisely on the antiviral activity of the active samples. In previous works, we documented the antiviral activity of Lippia sp. essential oils and some terpenes against dengue viruses, which was measured using more specific methods (plaque reduction assay and NS1 reduction assay) .We have cited these works in Discussion section (first and second version of the manuscript).

Comment 8. Results

“Figure 1: It could be included in the supplementary material”.

We have included the Figure 1 in the supplementary material.

Comment 9. Supplemental material:

“The table presenting the chemical constitution of the essential oils seems erratic. The compounds are not presented in their elution order from a DB-5MS column as stated by the authors, however their retention indices are calculated on a DB-5MS column and it is confusing. Their order in the table should be changed according to their order of elution..”.

 We have rewritten the Table S1, the compounds are listed according to their order of elution.

Comment 10. Results

“Figures 2 and 3: They show the antiviral effect on one virus serotype. Why are the results from the second serotype not included in the figure, they should show some variations”. “The reference antiviral agent should be included in the tables and the charts for comparison”.

 There was no differences in the antiviral potential of the tested EOs with respect to virus serotype. As shown in Table 3, samples with strong, weak and inactive activity against DENV-1 were also in this way against DENV-2. In addition, the IC50 values (Table 3) of three EO samples were similar between serotypes; they were in the same range of antiviral potential. This is because we selected results of cytopathic effect (CPE)-based antiviral assays with DENV-2 but not DENV-1 to include in Figures 2 and 3. Variations of the antiviral effect of tested EOs with respect to virus serotypes were showed in Figure S1 (first version of the manuscript), which presents the reference antiviral agent (SDS, sodium dodecyl sulfate). However, we agree and follow the reviewer's suggestion. For clarity, we have included a figure in the text of the second version of the manuscript (Figure 1) that presents the antiviral activity of each essential oil against DENV-1 and DENV-2, and we have removed Figure S1 because it is redundant. In addition, we have updated Figures 2 and 3, the antiviral activity is expressed as strong, weak and inactive categories, as it refers to both serotypes (Table 3).

Comment 11. Results

“Table 2: The presentation of the grouped components in a separate table is redundant at that point, since the following heatmaps show clearly the distribution of the chemical groups in each sample. Instead, a table including the major compounds in order of elution would be more suitable….”

 We have rewritten the Table 2, the main constituents of each essential oil according to their order of elution are listed.

We have included percentages of the terpene types in the heat maps in Figure 2 and 3, because they are mentioned in the text when making comparisons between essential oils, and they were used in the cluster analysis.

Comment 12.

“The text requires extensive linquistic improvement and needs to be checked for some typos also, such as line 451: seeded in, Table 3. basilicum.”

 We have rewritten some section of the text

Reviewer 3 Report

The manuscript "Study of essential oils from Colombian plants as potential sources of dengue herbal medicines based on chemical composition, in vitro and in silico antiviral analyses" deals with the extraction of fourteen EOs samples from six plants used in Colombian folk medicine by distillation. 

The work is accurate and well organized. However, some minor revisions are required, as follows:

The state of the art on the extraction of essential oils from vegetable plants can be enlarged, adding some recent works, as the ones of Baldino and Reverchon, De la Ossa et al., Mainar et al., etc., in order to highlight the possibility of performing this operation in a green way (i.e., using supercritical CO2) and to discuss the effect of the purity of EOs on their medical performance.

Check journal template.

Author Response

Manuscript ID: molecules-1919658

Authors: Lina Silva-Trujillo, Elizabeth Quintero-Rueda, Elena E Stashenko, Sergio Conde-Ocazionez, Paola Rondón-Villarreal, Raquel E Ocazionez *

Responses to Reviewer 3

 We thank the reviewer for his/her valuable suggestions and for recognizing the value of the presented data. We present here our replay to the comment.

Comment:

“The work is accurate and well organized. However, some minor revisions are required, as follows:

The state of the art on the extraction of essential oils from vegetable plants can beenlarged, adding some recent works, as the ones of Baldino and Reverchon, De la Ossaet al., Mainar et al., etc., in order to highlight the possibility of performing this operation ina green way (i.e., using supercritical CO2) and to discuss the effect of the purity of EOs ontheir medical performance..”

 We made an extensive review of the manuscript.

We are very interested in the green approach to perform various processes. In fact, several of our research lines are related to the implementation of circular bioeconomy in processing aromatic plants towards essential oils and extracts. We are currently examining the bioactivity of extracts obtained from several aromatic plants using hydroethanolic mixtures and supercritical carbon dioxide. However, this manuscript deals exclusively with essential oils. We must abide to current international standards regarding isolation techniques. ISO 9235.2021 contains definitions of several natural raw materials. The text for essential oils in this standard is: "product obtained from a natural raw material (3.20) of plant origin, by steam distillation, by mechanical processes from the epicarp of citrus fruits or by dry distillation, after separation of the aqueous phase – if any – by physical processes". If no water vapor is employed to separate the substances from the plant material, the resultant mixture should not be called essential oil. Similarly, according to the French Agency for Normalization (NF T 75-006), "The essential oil is the product obtained from a vegetable raw material, either by steam distillation or by mechanical processes from the epicarp of Citrus, or 'dry' distillation". Therefore, the essential oils studied in this work were obtained with water vapor (hydro distillation).

Round 2

Reviewer 2 Report

The manuscript under study has been considerably improved and the authors have responded to all issues addressed.

i.                     The written language has been sufficiently improved, however, there are still a few typos needed to be checked. Line 44: treating DENV; line 48: to treat human; line 74: is used; line 88: “species” not “specie”; line 242: monoterpene and sesquiterpene types; line 402: monotepene alcohols.

ii.                   Table S1: α-thujene not -tujene. Table S1 could also include the section of grouped compounds.

iii.                 CPE assay is undoubtedly a reliable method for preliminary analysis of antiviral effects. The additional employment of real-time PCR was a suggestion to further investigate and confirm the effect of specific active samples and/or promising compounds.